# Scale free topology as an effective feedback system

**Alexander Rivkind**[1,2☯¤], **Hallel Schreier**[3,2☯], **Naama Brenner**[4,2]*, **Omri Barak**[1,2]

**1** Rappaport Faculty of Medicine, Technion – Israel Institute of Technology, Haifa, Israel, **2** Network Biology Research Laboratories, Technion – Israel Institute of Technology, Haifa, Israel, **3** Interdisciplinary Program in Applied Mathematics, Technion – Israel Institute of Technology, Haifa, Israel, **4** Faculty of Chemical Engineering, Technion – Israel Institute of Technology, Haifa, Israel

☯ These authors contributed equally to this work.
¤ Current address: Department of Neurobiology, Weizmann Institute of Science, Rehovot, Israel
* nbrenner@technion.ac.il

**Data Availability Statement:** All relevant data are posted on a github webpage https://github.com/sashkarivkind/hallelNet.

**Funding:** This work was supported in part by the Israeli Science Foundation (isf.org.il grant number 346/16, OB; and grant number 155/18, NB). The

## Abstract

Biological networks are often heterogeneous in their connectivity pattern, with degree distributions featuring a heavy tail of highly connected hubs. The implications of this heterogeneity on dynamical properties are a topic of much interest. Here we show that interpreting topology as a feedback circuit can provide novel insights on dynamics. Based on the observation that in finite networks a small number of hubs have a disproportionate effect on the entire system, we construct an approximation by lumping these nodes into a single effective hub, which acts as a feedback loop with the rest of the nodes. We use this approximation to study dynamics of networks with scale-free degree distributions, focusing on their probability of convergence to fixed points. We find that the approximation preserves convergence statistics over a wide range of settings. Our mapping provides a parametrization of scale free topology which is predictive at the ensemble level and also retains properties of individual realizations. Specifically, outgoing hubs have an organizing role that can drive the network to convergence, in analogy to suppression of chaos by an external drive. In contrast, incoming hubs have no such property, resulting in a marked difference between the behavior of networks with outgoing vs. incoming scale free degree distribution. Combining feedback analysis with mean field theory predicts a transition between convergent and divergent dynamics which is corroborated by numerical simulations. Furthermore, they highlight the effect of a handful of outlying hubs, rather than of the connectivity distribution law as a whole, on network dynamics.

## Author summary

Nature abounds with complex networks of interacting elements—from the proteins in our cells, through neural networks in our brains, to species interacting in ecosystems. In all of these fields, the relation between network structure and dynamics is an important research question. A recurring feature of natural networks is their heterogeneous structure: individual elements exhibit a huge diversity of connectivity patterns, which

funders had no role in study design, data collection and analysis, decision to publish, or preparation of the manuscript.

**Competing interests:** The authors have declared that no competing interests exist.

complicates the understanding of network dynamics. To address this problem, we devised a simplified approximation for complex structured networks which captures their dynamical properties. Separating out the largest "hubs"—a small number of nodes with disproportionately high connectivity—we represent them by a single node linked to the rest of the network. This enables us to borrow concepts from control theory, where a system's output is linked back to itself forming a feedback loop. In this analogy, hubs in heterogeneous networks implement a feedback circuit with the rest of the network. The analogy reveals how these hubs can coordinate the network and drive it more easily towards stable states. Our approach enables analyzing dynamical properties of heterogeneous networks, which is difficult to achieve with existing techniques. It is potentially applicable to many fields where heterogeneous networks are important.

## Introduction

Complex networks, their structure and dynamics, have become an indispensable tool for investigating biological systems [1–3]. Networks provide a mathematical model for a system of many interacting components; such models are a cornerstone of computational neuroscience and are widely used in cell biology to describe genetic networks, protein interactions, metabolism and more. Motivated at least partly by these applications, the mathematical theory of network analysis has gained general and fundamental interest and has advanced tremendously in the past decades. Topics of interest include network structure and topology; dynamic behaviour such as fixed points, their stability and their scaling with network size; robustness of these dynamic properties to noise and to evolution; and network controllability and information processing properties [4–7].

One important property of biological networks that has raised much interest is their heterogeneous topology. Analyses of metabolic, protein interaction, gene regulatory and neural networks all show heterogeneous connectivity, including heavy tail distributions and modular structure [8–14]. Heterogeneous networks, such as those with a broad connectivity distribution, are generally more difficult to analyze; inferring their detailed topology requires exceedingly high statistics. Power-law distributions often provide a good approximation to such networks, but statistical difficulties have led to some debate concerning their adequacy [15].

Understanding how heterogeneous connectivity profiles relate to properties of the associated dynamical system is a non-trivial theoretical challenge [16–20]. In particular, heterogeneity makes these networks non-tractable by mean field methods [20]. In the current work we propose a different view on this problem, which interprets network topology as an effective feedback system. We develop a novel approximation for network ensembles with a broad connectivity distribution, based on the dominant role of a handful of hubs in a finite network. Identifying such hubs [21], understanding their impact on network functionality [22], and advancing the related theory [23] are all active research areas. Approximating a few hubs at the tail-end of the distribution by a single effective hub, we map the problem of heterogeneous network dynamics to that of a homogeneous network coupled to a single external node: each scale free network is approximated by a single *lumped-hub* connected to a homogeneous network of the remaining nodes (the bulk). Despite the simplification, this mapping retains properties of the original network ensemble. The new network is parameterized by hub strength and bulk connectivity.

We focus on the probability of random network ensembles to converge to attractors, and how this probability is modulated by ensemble topology. Convergence to fixed points is one of

the simplest dynamical behaviors, and hence a suitable starting point for analysis. Moreover, stable fixed points have traditionally been linked to fundamental functional properties of networks [24–26].

The lumped-hub approximation captures much of the behavior of heterogeneous network ensembles in terms of their probability of convergence to fixed points. The simplified ensemble can be analyzed by a combination of mean field approximations and feedback analysis. For networks with outgoing hubs, this analysis predicts a phase transition between converging and diverging dynamics as a function of the balance between hub strength and bulk connectivity, which is verified by numerical simulations. We show that the prediction holds on average for scale-free networks in a range of parameters relevant to real networks. Deviations from the theory stemming from large variability in scale-free ensembles as well as corrections to the mean field assumption are discussed.

Our results offer a novel framework for analyzing heterogeneous networks, which links insights from two different angles—internal dynamics of complex networks and the response to external input of simpler networks. The success of our mapping suggests that the role of outgoing hubs, in remarkable contrast to incoming ones [27], can be considered analogous to that of an external input in overcoming recurrent activity, suppressing chaotic dynamics, and ultimately driving the system to a stable fixed point. It highlights the crucial role of a small number of hubs in a finite network: their existence and relative strength is predictive of a given network's dynamics, possibly more so than other quantitative features of the distribution from which the connectivity was sampled.

## Results

### Dynamics with scale-free networks: Motivating observations

In what follows we analyze a model of nonlinear dynamics, often used to describe biological interactions such as in neuronal or genetic regulatory networks [24, 28]. Binary dynamic variables, $s_i$ ($1 \leq i \leq N$), approximate the activity of $N$ individual elements as being in one of two states: expressed (+1) or repressed (−1) genes, firing or quiescent neurons. The effect of elements on one another is described by a weighted sum of all incoming links:

$$s(t + 1) = \text{sign}(Ws(t)) \qquad (1)$$

where $W$ is the $N \times N$ connectivity matrix with real entries giving the strengths of interactions, drawn from some random network ensemble, and

$$\text{sign}(x) = \begin{cases} 1, & \text{if } x > 0 \\ 0, & \text{if } x = 0 \\ -1 & \text{if } x < 0. \end{cases} \qquad (2)$$

This is a special case of the widely studied Boolean networks [24, 29, 30], with the Boolean function chosen here to be the sign of the inputs' weighted sum (for significance and alternative definitions of the sign function, see Materials and methods).

The ensemble from which $W$ is drawn is a crucial ingredient of Eq (1). To study the effects of topology on the dynamics, we define $W = T \circ J$ as a Hadamard (element-wise) product between a random topology matrix $T$ and a random interaction strength matrix $J$. The 0/1 adjacency matrix $T$ defines a directed graph, whose edges are sampled from specified distributions of incoming and outgoing connections (see Materials and methods). The strengths $J$ are i.i.d. Gaussian variables with zero mean and standard deviation of unity.

For networks described by a directed graph, in general, incoming and outgoing degree distributions, $P_{in}(k)$, $P_{out}(k)$, need not be the same. In one case of special interest, gene regulatory networks, an empirical observation of such dissimilarity was reported. Outgoing connections were found to have a broad distribution consistent with a power-law: $P_{out}(k) \sim k^{-\gamma}$, a scale-free (SF) distribution, while incoming connections were much more narrowly distributed around the average [31]. These statistical properties represent the biological observation that while any given gene is regulated by at most a handful of others, some transcription factors can regulate the expression of up to hundreds of genes in the cell. These 'master regulators' are of much interest in cell biology, and have been at the focus of many specific molecular studies [32]. In neuroscience, evidence was also found for a hierarchy of connectivity, with scale-free topology suggested in several contexts [10, 11]. In particular, "leader neurons" were identified and studied [33].

In a recent work describing gene regulatory networks by random interaction matrices, it was found that among various network topologies, those drawn from ensembles with outgoing hubs have a high probability to converge to attractors under exploratory adaptation [27]. Ensembles that showed efficient ability for adaptation also exhibited a high probability of convergence to a fixed point attractor in their intrinsic dynamics.

Particularly noteworthy in this context is the observation that for the transpose random network ensemble, where incoming connections are broadly distributed whereas outgoing connection are not, the abundance of fixed point attractors is decreased dramatically. This result is illustrated for the model of Eq (1) in Fig 1A (open circles). For each of the two ensembles—Scale-Free Out degree distribution (SFO) and the transposed SFI—the fraction of simulations converging to a fixed point after a given time interval was computed; this is used as an estimate of the convergence probability across the ensemble. The result is presented as a function of network size $N$ for both ensembles. For SFO networks a considerable convergence

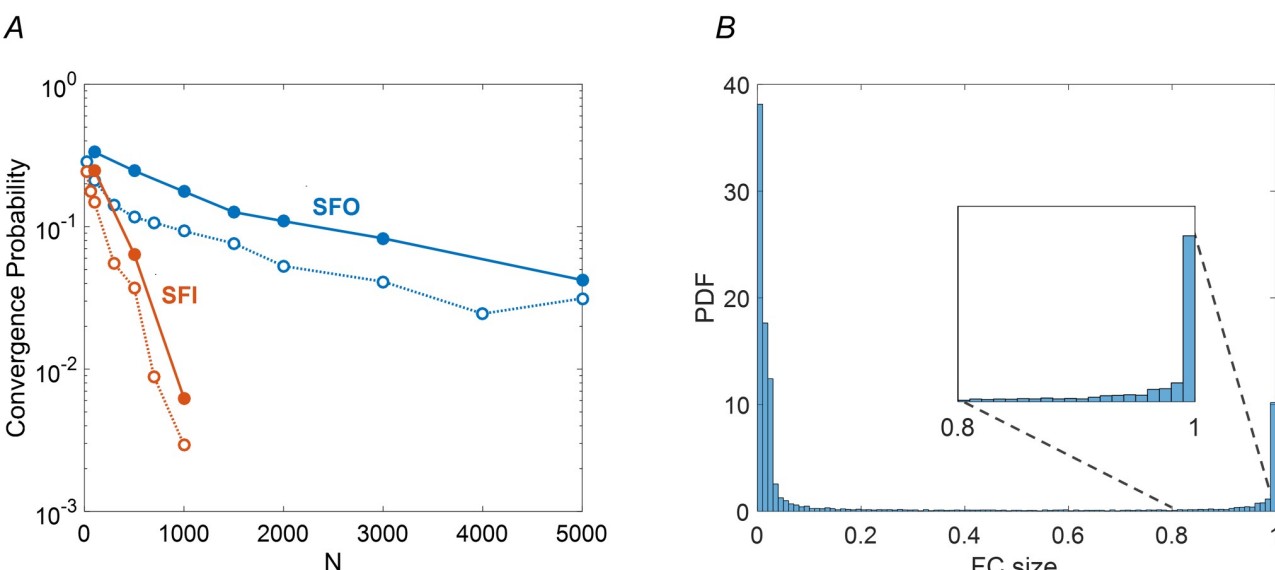

**Fig 1. Dynamic properties of Scale-Free-Out (SFO, blue) vs. Scale-Free-In (SFI, red) network ensembles. (A)** Convergence probability is shown as a function of network size $N$. Filled circles show convergence to frozen cores larger than 90% of the network; open circles show convergence to fixed points. Each data point represents an ensemble average over 600 networks with random $T$, $J$ and initial conditions. **(B)** Distribution of frozen core sizes (FC) relative to network size, across the SFO ensemble for $N$ = 1500. The distribution is further zoomed near FC = 1 justifying the notion of quasi fixed point, defined in the text and elaborated in Materials and methods. Scale free parameters for both panels are $\gamma$ = 2.2 and $k_{min}$ = 1, Binomial parameters are $N$ and $p = k/N$ with $k \sim 5$ (exact value of $k$ is determined by the realization of scale free distribution). See Materials and methods for details).

probability is maintained up to the maximal network size tested, $N = 5000$, whereas for SFI networks this probability decreases rapidly and is negligible for networks with $N > 1000$. These results are consistent with those of [27], where a model with continuous dynamics was used, extending the observation on the two transposed ensembles to Boolean dynamics.

When simulating these Boolean dynamics, we noticed that networks often converge to a state where the vast majority of the nodes are fixed—"frozen", while the rest continue to change (Fig 1B). From a biological perspective these are of interest as partially fixed states, and mathematically they have been studied in the context of general Boolean networks [30]. In what follows we define a quasi-fixed-point (QFP) as a state in which most ($> 90\%$) of the nodes are frozen. Fig 1A (filled circles) demonstrates that the probability of convergence to a QFP behaves qualitatively similar to the corresponding probability for fixed points: the remarkable difference between SFO and SFI ensembles and its dependence on network size remain the same. This shows that the constraining effect of outgoing hubs on network dynamics is not a unique feature of fixed points, but is also present for weaker notions of convergence.

Our empirical observations rely on convergence of dynamics—and hence is affected both by the existence of fixed points and by their stability. In Materials and methods we show that networks from both SFO and SFI ensembles typically have $O(1)$ fixed points. This suggests that the reported differences between ensembles stem from stability of fixed points. In the following sections we develop an approach which enables to better understand the stability properties of the two ensembles.

## The lumped-hub approximation

A finite realization of a power-law connectivity distribution is dominated by a handful of hubs connected to a macroscopic fraction of the network. Fig 2A shows an empirical histogram of the outgoing connections in an SFO network with $N = 1500$, where the largest hubs each connect to $\sim 1000$ nodes. To capture the properties of such a network we divide it into a small group of leading hubs, and the "bulk"—the rest of the network. The $m$ largest hubs are lumped into one node, while the remaining nodes are substituted by a homogeneous network with binomially distributed incoming and outgoing degrees, preserving the average connectivity.

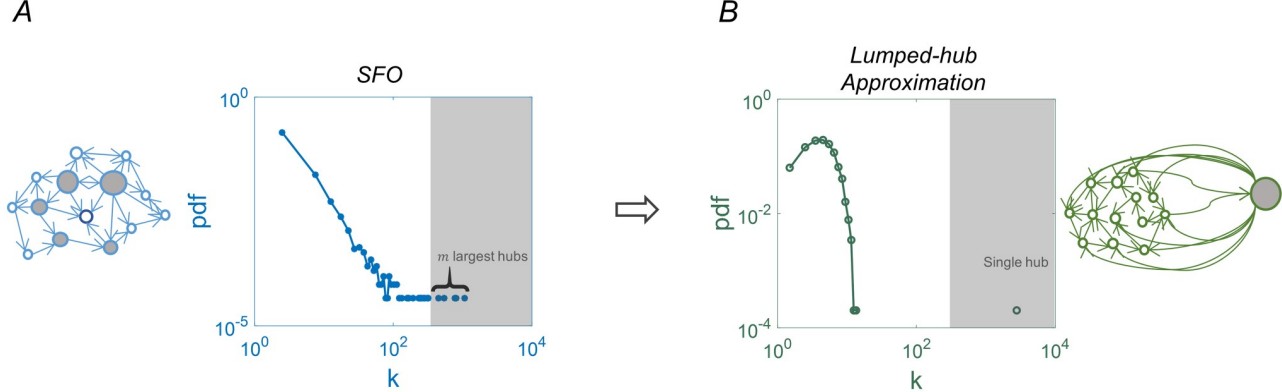

**Fig 2. The lumped-hub approximation.** (A) In a finite network with SFO degree distribution, the $m$ largest hubs (gray shading) are connected to a macroscopic fraction of the network. The histogram shows the number of outgoing connections from each node. (B) In the lumped-hub approximation of the same network, these are substituted by one effective hub and the rest of the nodes are approximated by a Binomial network. The connection strengths of the single hub to the bulk are determined by summing the strengths of the $m$ lumped hubs. The mean degree of the bulk, $k_b$, is retained in the mapping. Here $N = 1500$.

We call this mapping the "lumped hub approximation"; the resulting network is characterized by the bulk mean connectivity $k_b$ and the pattern of connections $u_i$ from the lumped-hub to each node $i$ in the bulk. These depend both on the original network topology and the lumping parameter $m$.

The connections $u_i$ are derived by summing the contributions of all the hubs that were lumped according to:

$$u_i \sim \mathcal{N}(0, \sum_{j=1}^{m} T_{ij})$$

(3)

where we assume that hubs in the original SFO network are represented by rows $1, \ldots, m$ of the connectivity matrix. In this construction, if more than one of the largest $m$ hubs in the original network were connected to node $i$, then the connection between the lumped hub and this node is drawn from a distribution with appropriately scaled variance. We refer to this setting as *Exact Pattern* approximation, since the connectivity from the lumped hubs is fully preserved (as opposed to coarser approximations discussed below).

Despite the crudeness of the lumping approximation, we find that it preserves dynamic properties of scale free networks at the ensemble level. Fig 3 shows that the probability to converge to a QFP in the approximated networks follows the same dependence on network size as the original scale free ensembles. In particular, the significant difference between SFO and SFI, mapped to either an outgoing or incoming lumped hub, is captured by the approximation. These results are presented for a specific SFO ensemble, characterized by a fixed set of parameters (see figure caption for details).

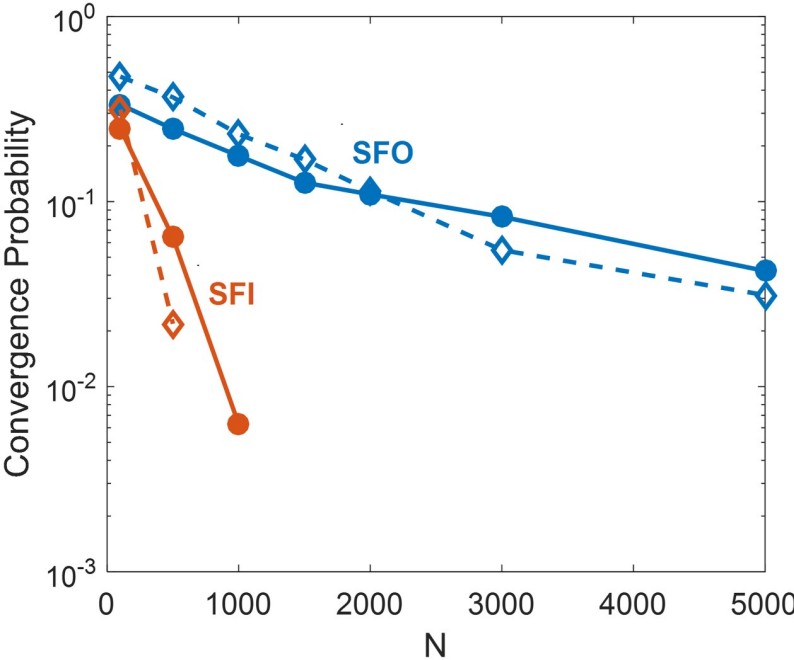

**Fig 3. Convergence probability of SFO/SFI lumped-hub approximation.** Probability of convergence to a Quasi Fixed Point (QFP), for SFO and SFI (blue and red filled circles respectively; same data as in Fig 1A). Their corresponding lumped-hub approximations are shown in blue and red open symbols with lumping parameter $m = 4$. Statistics and parameters as in Fig 1.

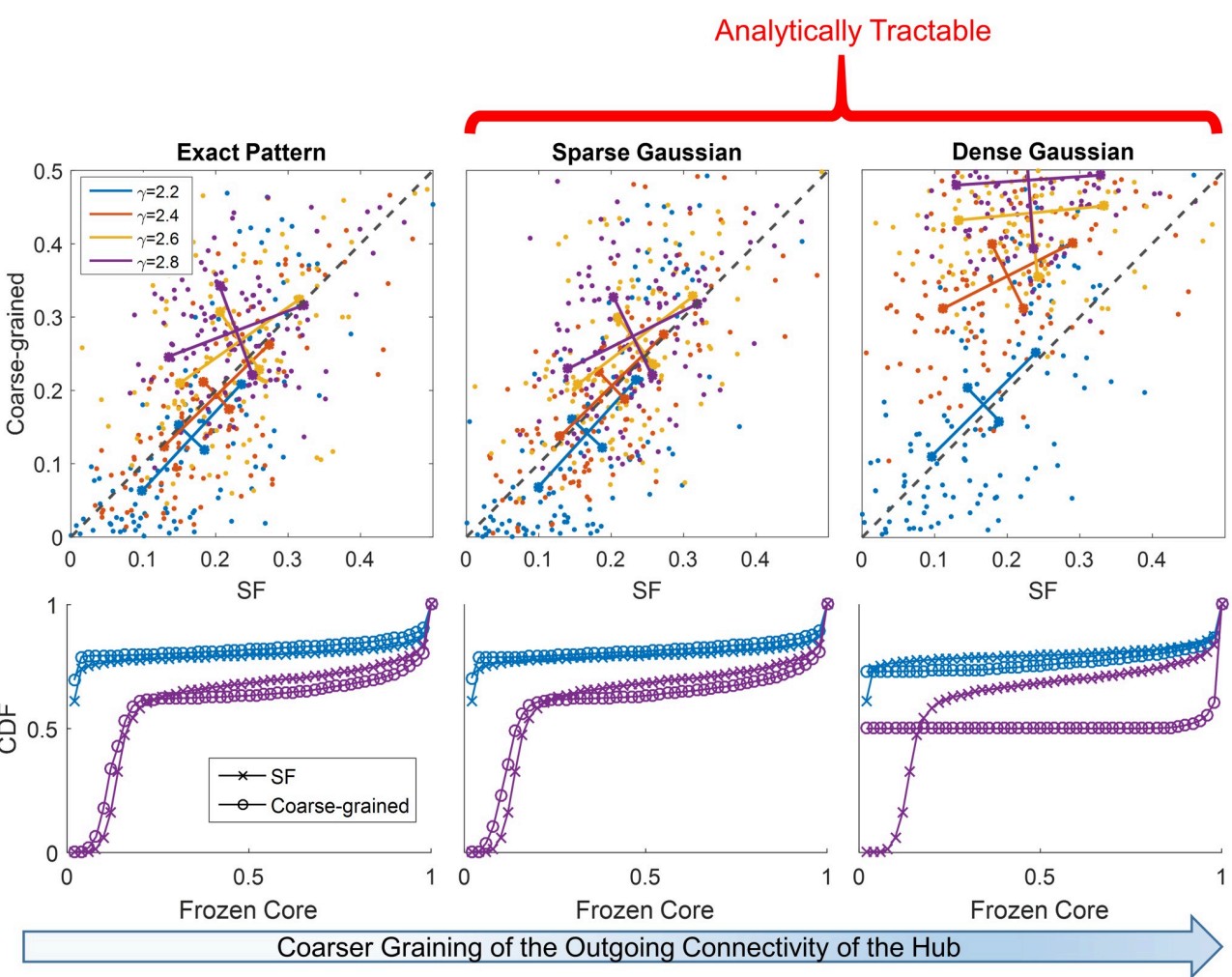

**Fig 4. Lumped-hub approximation captures SFO network properties for a range of power-laws.** Within the lumped-hub approximation, we show three coarse grained versions of the hub outgoing connectivity pattern. **(Left)** Exact pattern of connectivity is maintained. **(Middle)** Number of nonzero connections is maintained, but their weights are replaced by a Gaussian distribution. **(Right)** A single Gaussian distribution describes all outgoing connections. **(Top)** Probability of converging to QFP for SFO networks and their respective lumped-hub approximations with increasingly coarse graining. For each SFO parameter $\gamma$ (color code and legend), 125 topology realizations $T$ were drawn, and the convergence probability was estimated from 50 realizations of the connection strength $J$ and initial conditions. On each colored cluster of points, longer lines denote the direction of the linear fit to each $\gamma$, and the size of the perpendicular segment is the remaining variance after the fit. The identity line is added for reference (dashed black). Data points were jittered by a small uniform random noise (in range of ±0.01) to avoid their overlap due to discrete sampling. **(Bottom)** Cumulative distributions of frozen core sizes for SFO networks and their lumped-hub approximations for two power-law exponents: $\gamma$ = 2.2, 2.8 (color coded as in the top panels). All scale free distributions have $k_{min}$ = 1 (see Materials and methods for more details).

We next ask whether our lumping approximation can reproduce the range of different behaviors across SFO ensembles with various power-law distributions. To answer this question we map a broad range of SFO ensembles to lumped-hub networks and compare their dynamic properties. Fig 4 (top left) shows a scatter-plot of convergence probabilities for SFO networks vs. their lumped-hub approximations. Each point represents a specific SFO topology realization $T$, and the convergence probability is estimated as an average over realizations of connection strength $J$. It is seen that the convergence probability is largely retained in the mapping across a range of scale free parameter $\gamma$. In addition, the bottom-left panel of Fig 4 shows that the distribution of frozen core (FC) sizes, exhibiting a bi-modal shape dependent on ensemble parameters, is also very well captured by the lumping approximation. Comparisons of the

autocorrelation and dimensionality of non-converging trajectories are given in Materials and methods, also supporting the similarity of the approximation to the original networks. These results demonstrate that our lumped-hub approximation describes dynamic properties of the SFO network ensembles across a broad range of parameters.

## Feedback analysis for the lumped-hub network ensemble

We have seen that despite its coarseness, the lumped hub approximation somehow captures an important feature of scale free networks that strongly affects their dynamics. Can it be harnessed to gain a deeper understanding of these networks?

In particular we are interested in the significantly larger convergence probability in the SFO ensemble compared to the corresponding SFI. The essence of this difference becomes intuitively clear in the lumped-hub approximation. In lumped SFO networks, the hub coherently drives a macroscopic fraction of the bulk nodes. In contrast, in lumped SFI networks the hub receives a large number of inputs but only drives a small fraction ($\approx k_b$) of the nodes. Previous work showed that intrinsically chaotic dynamics of homogeneous random networks can be suppressed by a coherent external drive [34]. Although here the hub is not external to the network, we argue that a similar suppression effect underlies convergence in SFO networks.

The lumped-hub network is described by two coupled equations of the bulk and the hub:

$$s'(t + 1) = \text{sign}(W's'(t) + uh(t)) \tag{4}$$

$$h(t + 1) = \text{sign}(v^T s'(t)) \tag{5}$$

where $s'$ and $W'$ denote the bulk activity and connectivity respectively, the scalar $h$ is the hub activity and the vectors $u$, $v$ are the connections between the bulk and the lumped hub.

Our results so far relied on the outgoing connectivity from the lumped hub maintaining the *Exact Pattern* of the original connectivity, according to Eq (3). To facilitate the analysis, we consider an ensemble that retains statistics of the lumped hub connectivity rather than its exact pattern:

$$u_i \sim \begin{cases} \mathcal{N}(0, \sigma_h^2), & \text{with Prob } \alpha \\ 0, & \text{with Prob } (1 - \alpha) \end{cases} . \tag{6}$$

This ensemble is characterized by three parameters: mean degree of the bulk $k_b$; the hub sparseness $\alpha$, representing the fraction of its nonzero outgoing connections; and the variance $\sigma_h^2$ representing its connection strengths. Assuming a Gaussian distribution of connections, this defines the *Sparse Gaussian* ensemble (Fig 4, center). This approximation is shown to predict both convergence probabilities (top) and frozen core statistics (bottom) nearly as well as the lumped hub with the *Exact Pattern* of connectivity.

Notably, a coarser model, which we refer to as the *Dense Gaussian* approximation, and where sparseness is not preserved, turns out to be dramatically less accurate. This ensemble, where the hub is connected to the entire network by outgoing connections drawn from a Gaussian distribution with variance $\alpha\sigma_h^2$, tends to systematically overestimate convergence probabilities (Fig 4, right). In what follows, we therefore analyze the Sparse Gaussian ensemble to assess the influence of the lumped hub on network dynamics.

Consider first an auxiliary setting—the "open loop" setting—where the hub is held at a fixed value. Here the conditions on column mean [20] are fulfilled and Mean Field analysis can be applied. Effectively, connections from the bulk of the network to the hub are discarded, and it remains connected only through its outgoing links (Fig 5A). The hub then acts as an

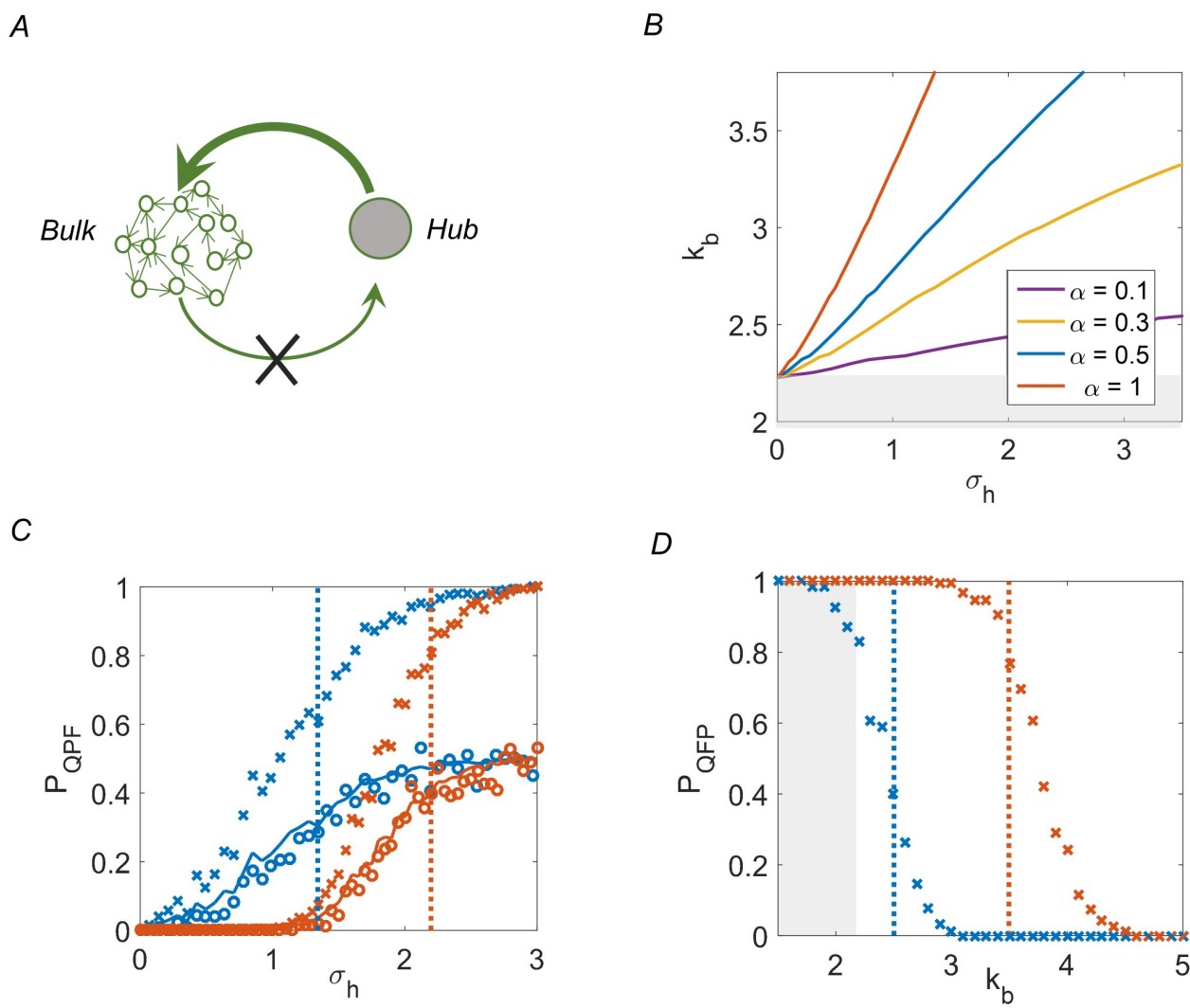

**Fig 5. Feedback analysis for the lumped-hub network ensemble.** (A) Conceptual scheme of feedback analysis. The lumped-hub network contains a single hub (gray) connected to a homogeneous network representing the bulk (green). In the open loop setting, the hub is clamped to a fixed value while its incoming connections from the bulk are disabled (×). (B) Mean Field Theory predicts a threshold value of $\sigma_h$ above which chaos in the bulk network is suppressed and the system is driven to a stable fixed point. These transition lines are plotted in the $(k_b, \sigma_h)$ plane for various values of hub sparseness $\alpha$. (C) The Probability of converging to a QFP for lumped-hub network ensembles computed as a function of hub strength $\sigma_h$, for open-loop (crossmarks, converging to 1) and closed-loop (circles, converging to 1/2). Two values of average bulk connectivity and sparseness are shown: $k_b = 3$, $\alpha = 0.5$ (blue) and $k_b = 5$, $\alpha = 1$ (red). MFT-predicted transition is depicted by vertical lines of the corresponding color. The solid lines are the open loop values divided by two. The numerical results confirm both the transition point and the factor of *half* due to closing the loop. $N = 1500$, and statistics is obtained from 239 realizations. (D) Same as (C), for the open loop case and as a function of $k_b$ for fixed $(\sigma_h, \alpha)$ with $N = 5000$. Blue: $(\sigma_h = 1.0, \alpha = 0.25)$, red: $(\sigma_h = 2.12, \alpha = 0.5)$. Statistics is obtained from 180 realizations. Gray regions in (B), (D) denote the limit of validity of MFT (See section *Limits to the lumped hub approximation*).

input to the rest of the network, a simple situation that can be analyzed by Mean Field Theory. Formally, we replace the dynamic variable $h$ by a fixed value (+ 1 without loss of generality), namely Eq (5) is substituted by $h(t + 1) = 1$. In this open loop setting, the regularizing effect of the fixed input competes with the recurrent dynamics of the bulk, and we ask under what conditions the system converges to a fixed point. Later, we relate the open-loop results back to a fully recurrent network including the hub as a dynamic variable [35].

To determine whether or not the system converges to a fixed point under input, we consider the time-evolution of the Hamming distance $d(t)$ between two bulk trajectories $s'_1(t)$ and

$s_2'(t)$. In the thermodynamic limit $N \to \infty$, this evolution is deterministic and given by some function $f(d)$ (see Ref. [36] and Materials and methods for derivations):

$$d(t + 1) = f(d(t)). \tag{7}$$

The external drive $h$ can suppress chaotic activity and drive the network to a stable state $s'^*$ if and only if $d^* = 0$ is a stable fixed point of the mapping Eq (7). This, in turn, requires the derivative of $d$ at the origin, denoted by $\nabla d$, to be smaller than *one*. We show in Materials and Methods that, within Mean Field Theory (MFT), the condition for this can be expressed as a threshold on the hub strength,

$$\sigma_h > \sigma_{crit}(k_b, \alpha) \tag{8}$$

with $\sigma_{crit}$ depending on the average bulk connectivity $k_b$ and the sparseness $\alpha$. Fig 5B shows this critical line for several values of $\alpha$ in the $(\sigma_h, k_b)$ plane. Importantly, the lines are distinct for different hub sparsity $\alpha$. As already seen in Fig 4 (right), ignoring sparsity leads to an oversimplified and inaccurate model.

Fig 5C illustrates the accuracy of our theory. Blue and red × symbols show simulated probability of convergence rising from zero to one as a function of $\sigma_h$ for two sets of $(k_b, \alpha)$, with the critical values predicted by Eq (8) (vertical lines in corresponding colors). These results demonstrate both the qualitative prediction—a (smoothed) transition from zero to one in convergence probability, and the correct location of the transition at the predicted $\sigma_{crit}$. The width of the transition is affected by two factors—the slope of the critical lines in the $(\sigma_h, k_b)$ plane (Fig 5B) and the finite size of the system. We demonstrate these effects by using a larger network and traversing the plane along $k_b$ instead of $\sigma_h$, resulting in a much sharper transition (Fig 5D). We therefore conclude that MFT adequately predicts the open-loop convergence probability. Note that the critical driving strength increases indefinitely for large $k_b$, in agreement with previous results [36, 37].

Returning now to the closed-loop network dynamics Eqs (4) and (5), where feedback to the hub is restored and it is a dynamic variable with incoming and outgoing connections, we ask whether a stable attractor under external drive is consistently maintained as an attractor of the recurrent dynamics. Intuitively one may argue that, with probability 1/2 with respect to the random connections in the ensemble, the clamped value will be consistent with the hub's input; the driven steady-state will then also be an attractor of the full recurrent dynamics. This intuitive argument only refers to *existence* of a closed-loop fixed point, and does not guarantee its stability. Therefore, beyond the open-loop stability criterion Eq (8), the closed-loop setting requires to consider the recurrent dynamics of the hub. As long as the hub does not flip from its clamped value ($h$ changes from +1 to −1), open and closed loop system dynamics coincide.

Consider a state $s'$, a small distance $d = \delta > 0$ away from the fixed point $s'^*$. For a stable fixed point we expect exponential convergence with a rate $\nabla d$. Taking into account the finite size of the network, we estimate that convergence to a distance of less than one node ($d < N^{-1}$) takes approximately

$$T_{conv} \approx -\frac{\log(N\delta)}{\log \nabla d}. \tag{9}$$

Preserving this stability in the full closed-loop network requires that the hub does not flip its sign during the time of convergence:

$$T_{conv} < T_{flip}. \tag{10}$$

In Materials and Methods we show that the typical hub flipping time $T_{flip}$ is approximately:

$$T_{flip} \approx \frac{N}{\sqrt{k_h}\delta} \tag{11}$$

which implies that Eq (10) holds for $\nabla d$ far enough from the phase transition. Furthermore, it is argued that even in the proximity of the critical value $\nabla d \approx 1^-$, the corrections due to closing the loop are smaller than the error of the mean field approximation made in the open loop analysis. Taken together, these arguments show that all effects of closing the loop will reduce the probability of the system converging to a stable fixed point by half. Fig 5C shows this is indeed the case. Circles depict closed-loop simulations of lumped-hub networks, and the matching lines show the smoothed result for open-loop divided by two.

## Applicability of feedback analysis to finite SFO networks

We now return to evaluate the accuracy of our mean field approximation to the original scale free network ensembles. Making the connection between the theoretical results and empirical simulations entails two steps: first, for each scale free network realization we define the lumped-hub parameters $(k_b, \sigma_h, \alpha)$; second, these parameters are used to compute $\nabla d$ in MFT (Materials and Methods), predicting convergence for values smaller than one. Importantly, the first step is not a one-to-one transformation between the parameters of the scale free ensemble and those of the lumped-hub ensemble: due to the large heterogeneity of scale free networks, different realizations generally result in widely varying lumped-hub parameters and consequently in broadly distributed values of $\nabla d$. An example of this distribution is shown for one SFO ensemble in Fig 6A. In particular, lumped-hub parameters across the ensemble are such that $\nabla d$ is distributed on both sides of the phase transition (dashed vertical line, $\nabla d = 1$). This wide distribution for fixed generative parameters, implies a lack of sharp phase transition when these parametes are varied.

Pooling together realizations of different power-law parameters opens a broader range of $\nabla d$ such that comparison to the theory is more meaningful. We simulated many SFO networks with a range of underlying power-law parameters and tested the prediction of a transition as a function of $\nabla d$. Fig 6B compares the theoretical prediction for the convergence probability (dotted black line) with the pooled simulation results (blue symbols); despite the pooling of a highly variable set of simulations, $\nabla d$ is revealed as a crucial determinant of convergence. The smooth sigmoid obtained follows the transition around the predicted value of $\nabla d = 1$. A decrease in convergence probability is found for very low $\nabla d$ values, in contrast to the prediction; this effect will be discussed below. For one SFO ensemble, the simulations follow the prediction but cover only a small range of $\nabla d$ (red symbols).

Empirical convergence probabilities for scale free networks also depend on network size (Fig 1). This dependence can be partially explained by the lumped-hub approximation: for a given scale free ensemble, we can follow the lumped-hub parameters averaged over the ensemble as a function of network size $N$. Fig 6C shows that the average $(k_b, \sigma_h, \alpha)$ values progress towards the unstable regime as network size increases, with $k_b$ being influenced most strongly; these effects are consistent for several scale free ensembles (colors). The implications of this trend can be seen in the dependence of convergence probability on network size, shown in Fig 6D. Simulation results of scale free networks and their corresponding lumped-hub approximations are shown together with the theoretical prediction, showing that most of the $N$-dependency is captured correctly.

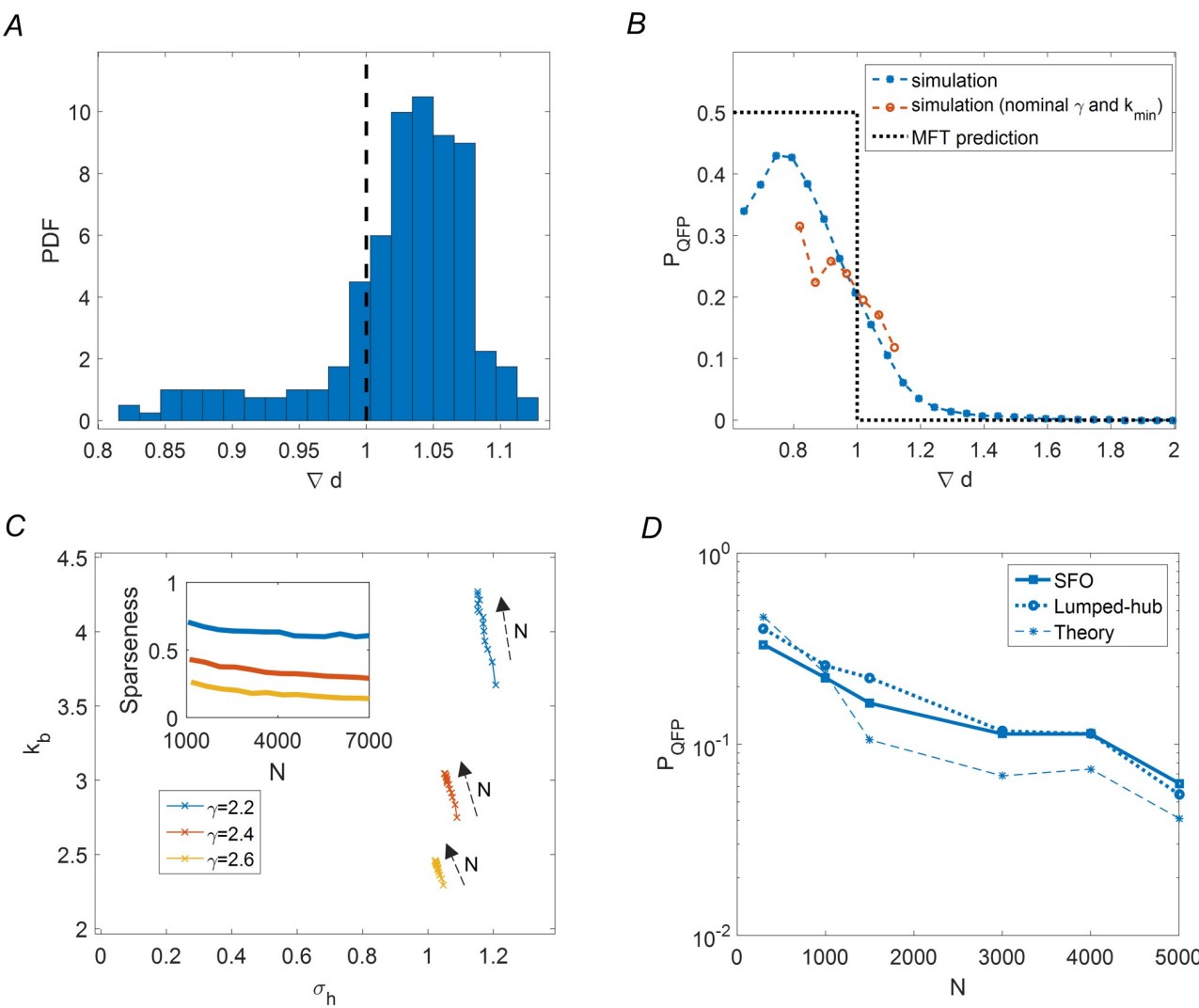

**Fig 6. Comparing lumped-hub MFT prediction with SFO simulations. (A)** Variability of lumped-hub parameters in a single scale free network ensemble: histogram shows the distribution of $\nabla d$ as computed from the lumped-hub parameters, for a scale free ensemble with $\gamma = 2.4$, $k_{min} = 1$. Black dashed line: MFT predicted phase transition, $\nabla d = 1$. **(B)** Transition in convergence probability $P_{QFP}$ (blue symbols) computed for networks with different $k_{min}$ values in range $0.5 \leq k_{min} \leq 2$ and $\gamma = 2.2, 2.4, 2.6$. For each set of parameters 256 network realizations of size $N = 1500$ were simulated. Results were then binned by $\nabla d$ width of 0.05. Bins with less than ten samples were omitted from the plot (restricting the standard error to 0.16 per point). A short red line shows the data for the single ensemble of panel A. **(C)** Effect of network size on average lumped hub parameters $k$, $\sigma_h$, and $\alpha$ (sparseness; inset). Increasing network size causes a gradual shift towards the unstable phase (see Fig 5B), with $k_b$ affected most strongly. Colors show different $\gamma$. **(D)** Effect of network size on convergence probability: comparing simulation, lumped-hub approximation and theory. Empirical data points represent 256 Monte-Carlo realizations for scale free with $\gamma = 2.4$, $k_{min} = 1$.

## Limits to the lumped hub approximation

The lumped hub approximation relies on the possibility of decomposing the network into a hub and bulk nodes. In turn, the phase transition predicted for lumped-hub networks relies on this decomposition and its relation to homogeneous networks driven by an external input. This approximation holds for a range of parameter values of practical interest. To understand the limits of validity of the approximation, we consider several conditions that are required to hold.

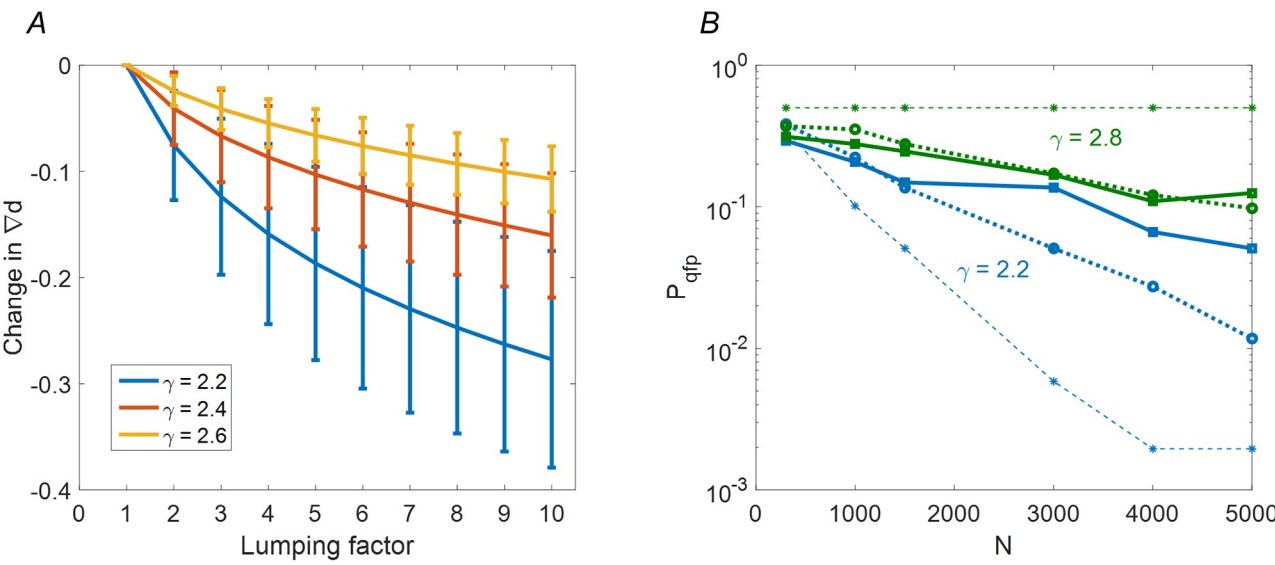

**Fig 7. Limits of the approximation. (A)** Sensitivity of lumped hub approximation to the number of leading nodes being lumped is shown for 1000 realisations of scale-free topology with fixed parameters $k_{min} = 1$, $N = 1500$, and for three values of $\gamma = 2.2, 2.4, 2.6$. For each realization, the $\nabla d$ at $m = 1$ is taken as reference, from which the subsequent decrease in $\nabla d$ is measured. **(B)** Fig 6D is replicated for $\gamma = 2.2, 2.8$. In the large $\gamma$ case, the lumped hub model holds, while mean field theory becomes invalid due to extreme sparseness. For low $\gamma$ it is the lumped hub model per se that becomes inaccurate, due to sensitivity to choice of $m$.

First, the most highly connected nodes are being lumped together to one effective hub. The approximation thus neglects their internal dynamics, which should therefore be of little importance for convergence of the system. Ideally there should exist an interval of lumping numbers $m$ across which $\nabla d$ remains largely unchanged. Fig 7A shows that this is the case for SFO networks with $\gamma \geq 2.4$ (a change of roughly 0.1 compared to a width of more than 0.2 in Fig 6B). For $\gamma = 2.2$ we see that $\nabla d$ depends more strongly on the choice of $m$, compromising the stability of the approximation. Indeed, comparing the approximation to direct simulation results in Fig 7B, shows a large discrepancy between scale free and lumped-hub behaviors for $\gamma = 2.2$.

Second, the network must remain *globally* connected. For very small mean connectivity, networks tend to split into a number of disconnected components. In this situation, the probability of an individual realization converging to a fixed point drops, while realizations with multiple fixed points emerge (See Materials and methods). We believe that this effect is responsible for the drop in convergence probability observed at extremely low $\nabla d$ (Fig 6B).

In addition, for the approximation to be relevant, the effective external drive by the hub must be strong enough. As indicated in Fig 5B, for a sparsely connected hub ($\alpha = 0.1$) the transition between convergent and chaotic regimes is almost independent of hub strength (the regime where the hub becomes irrelevant is marked by a gray area at Fig 5B and 5D). This can explain the failure of the theory for low $\alpha$ values, that arise for high $\gamma$ values, as exemplified for $\gamma = 2.8$ in Fig 7B.

Finally, by lumping $m > 1$ dominant hubs into a single node, we are limited to a single fixed point of the open-loop system (up to symmetry of the hub activation). If several combinations of the $m$ hubs' states can stabilize the open-loop system, the possibility of multiple fixed point pairs in a single realization emerges. According to [38] and our Materials and Methods, such a possibility should reduce the probability of convergence. This can partially explain the slight drop below the factor of 1/2 in Fig 6B.

## Discussion

Broad connectivity profiles are a ubiquitous feature of naturally occurring networks. Understanding the dynamics arising from such topology, however, remains a great challenge. Our main hypothesis in this work was that, for such networks, a small number of nodes has a disproportionate effect on the entire system dynamics. We suggest a simplified model—the lumped-hub approximation—that makes this hypothesis explicit by lumping these nodes into a single hub and replacing the rest of the network by a homogeneous bulk.

We showed that the lumped-hub approximation predicts the behavior of scale free networks at the ensemble level, explaining the change in convergence probability as a function of power-law exponent and network size. Furthermore, the approximation is also helpful at the single realization level, indicating that a large part of the variability between scale-free networks stems from the variability in their effective lumped-hub strength and sparseness, and in the average bulk connectivity. In this respect, our proposed three-parameter description of topology is more predictive than the generative scale free parameters.

Our analysis was based on scale-free connectivity, but should apply for other topologies with broadly distributed out degree, for which mean field does not converge even with high moment corrections [20]. In fact, our results indicate that the existence of a small number of hubs connected to a macroscopic fraction of the network nodes, rather than the precise shape of the distribution tail, is a crucial factor in shaping network dynamical properties. The existence of one such outgoing (but not incoming) hub was sufficient not only to induce increased convergence to stable attractors, but also to endow the ensemble with an extremely high heterogeneity among realizations.

The simplicity of the lumped-hub network ensemble, with a homogeneous bulk network and a single hub connected to a fraction $\alpha$ of other nodes, allows analytic treatment of the problem. Specifically we applied feedback analysis, based on interpreting the hub as an external drive to the bulk, and then devising a mean field theory to determine convergence of the system. We then closed the loop while requiring consistency, allowing us to analytically estimate the probability for convergence. This analysis explains simply and intuitively why networks with outgoing hubs, and not those with incoming hubs, tend to converge to fixed points. We showed that, perhaps surprisingly, it is the stability of fixed points, rather than their abundance that underlies this effect.

In homogeneous networks, the average activity is a natural quantity of interest and provides the basis for mean-field approximations [36, 37]. For heterogeneous networks a low-dimensional approximation is still desirable, but the choice is no longer obvious. If the network is composed of homogeneous sub-networks a version of mean-field theory can be applied [39, 40]. Alternatively, a degree-dependent mean-field approach can be developed [41, 42]. Weighted averages of activity, guided by topology, can also serve to reduce the dimensionality of the dynamics [18, 19]. Our approach can be thought of as splitting the network and then approximating each part as homogeneous. This allows on one hand to tailor the approximation to specific heterogeneous realizations, but on the other hand to treat the approximate problem with mean-field and feedback tools.

The analytic treatment was developed for binary dynamic variables, but the empirical convergence properties and their dependence on outgoing hubs are shared also for a continuous variable model [27]. Therefore it is plausible that these properties reflect more generally the ability of outgoing hubs to promote convergence and to stabilize network dynamics.

We note the comparison of our results with previous work on Boolean networks with a randomly chosen Boolean function at each node [43], which found that the phase transition did not differ between SFO and SFI networks. By specifying a threshold function in our model, the

effect of a hub becomes coherent among all the nodes that it influences. In contrast, when randomizing over Boolean functions, the effect of the hub has no such coherence; it is this coherence which promotes convergence in the model studied here.

From a biological viewpoint, our results illustrate how outgoing hubs—master regulators—can act as global coordinators of dynamics in gene regulatory networks. This role is expected to be realized under strong perturbations, in addition to the usual role of hubs as regulators of local specific circuits. Indeed, recent experimental work in bacteria suggests an organizing role for hubs in the emergence of gene expression patterns following strong rewiring perturbations [44, 45]. This idea, suggesting a dual role for the structure of gene regulation networks depending on conditions, awaits further experimental and theoretical investigation.

## Materials and methods

### Suppression of chaos in the open-loop setting

We consider a lumped hub network ensemble, where the effective hub is clamped at a fixed value and acts as an input to the rest of the nodes. It is connected to the network by a vector $u$, covering a fraction $\alpha$ of the network, with connection strengths drawn from a Gaussian distribution with zero mean and standard deviation $\sigma_h$. Formally, we determine whether the input suppresses chaos by following the time evolution of the normalized Hamming distance between two trajectories $s^1(t)$ and $s^2(t)$, denoted by $d(t)$. Suppose that $d(t_0) = 1/N$, and assume, without loss of generality, that the states of interest $s^1$, $s^2$ differ at the first bit: $s_1^1 \neq s_1^2$ while $s_j^1 = s_j^2$ for $2 \leq j \leq N$. At time $t_0 + 1$ the distance $d$ will be determined by the number of rows in $W$ for which the first element changes its sign:

$$d(t_0 + 1) = \frac{1}{N} \sum_{i=1}^{N} \theta \left( |W_{i1}| - \left| \sum_{j=2}^{N} W_{ij} s_j + u_i \right| \right). \tag{12}$$

Equivalently, one may consider the derivative at the origin of the normalized distance $d$, given by:

$$\nabla d = N \Pr \left\{ |W_{i1}| > \left| \sum_{j=2}^{N} W_{ij} s_j + u_i \right| \right\}, \tag{13}$$

with the condition for stability being $\nabla d < 1$. The inequality in curly brackets can be realized only if $W_{i1}$ is nonzero; moreover the other matrix element can be grouped according to the number of nonzero elements in each row. This grouping results in the following convenient way to rewrite the last equation:

$$N^{-1} \nabla d = P_1 \sum_{k'=0}^{N} P_2(k') P_3(k') \tag{14}$$

with terms $P_{1,2,3}$ defined as:

$$P_1 = \Pr \{ W_{i1} \neq 0 \} \tag{15}$$

$$P_2 = \Pr \{ \#\{ W_{ij} \neq 0 \} = k' | j > 1 \} \tag{16}$$

$$\begin{aligned} P_3 &= \alpha \Pr \left\{ |z_1| > \sqrt{k' + \sigma_h^2} |z_2| \right\} + \\ &+ (1 - \alpha) \Pr \left\{ |z_1| > \sqrt{k'} |z_2| \right\} \end{aligned} \tag{17}$$

Here $z_1$, $z_2$ are i.i.d. normal Gaussian variables, as follows from the mean field approximation, and $i$ is an arbitrary row. The terms $P_1$ and $P_2$ are determined by the network structure: $P_1$ is simply the sparsity of the Binomial network which makes up the bulk nodes,

$$P_1 = N^{-1}k_b, \tag{18}$$

and the probability of a row in the network having exactly $k'$ nonzero entries is:

$$P_2(k') = P_{\text{binom}}(k', N-1, N^{-1}k_b). \tag{19}$$

The only term which depends on the actual realization of weights is $P_3$. To evaluate it we note that for a positive scalar $\eta$:

$$\Pr\{|z_1| > \eta|z_2|\} = \frac{2}{\pi}\arctan\frac{1}{\eta}. \tag{20}$$

To derive this expression, one notes that in the $|z_1|$, $|z_2|$ plane, we are interested in the portion of probability mass that lies below the line $|z_2| = \frac{|z_1|}{\eta}$. Since the probability density in this plane only depends on the distance from the origin, we simply compute the angle between this line and the $|z_1|$ axis and Eq (20) follows.

Combining all the terms together, and assuming large $N$, we arrive at:

$$\nabla d = \frac{2k_b}{\pi}\sum_{k'=0}^{N}P_{\text{poisson}}(k', k_b)\left(\alpha\arctan\frac{1}{\sqrt{k'+\sigma_h^2}} + (1-\alpha)\arctan\frac{1}{\sqrt{k'}}\right), \tag{21}$$

For $\alpha = 1$ and large $k_b$, the sum over $k'$ can be approximated by its expected value, and tanh by its argument leading to:

$$\nabla d \approx \frac{2k_b}{\pi\sqrt{k+\sigma_h^2}} \approx \frac{2k_b}{\pi\sigma_h}. \tag{22}$$

This expression accounts for the asymptotically linear relation between $k$ and $\sigma$ at the phase transition curve in Fig 5B. Conversely, for sparse hubs with $\alpha < 1$ there exists a $k_{max}$ above which we have $\nabla d > 1$ for any $\sigma_h$.

## Quasi fixed points

Along with a possibility of converging to a stable fixed point another scenario exists where most network nodes are fixed but a small fraction is still toggling.

More specifically, a situation where all nodes except a small, interconnected clique are frozen becomes typical. We refer to such a regime as a quasi fixed point (QFP). A threshold to define a QFP was set empirically at a value of 0.9 where a rise in probability of frozen cores starts (Fig 1B).

## Fixed points are always there, regardless of topology

Defined by equation Eq (1), our system turns out to be a special case of the Theorem in Ref. [38] which states that the expectation $\mathbb{E}M$ of number of fixed points $M$ in random Boolean networks is *one*, subject to a condition on the distribution of random Boolean functions that holds in our setting. Specifically, to comply with [38] an ensemble of random Boolean functions $\phi_i(s)$ defining the network dynamics via $s_i(t+1) = \phi_i(s(t))$, must have a set of *neutral links*, the removal of which renders the network acyclic. Here a link $j \to i$ between a pair of

nodes is said to be neutral iff

$$\Pr\{\phi_i(s) = g(s_1, s_2..s_j..s_N)\} = \Pr\{\phi_i(s) = g(s_1, s_2..\bar{s}_j..s_N)\} \tag{23}$$

for any Boolean function $g$. In our case of a threshold function, the condition Eq (23) is fulfilled immediately because

$$\Pr\left\{\phi_i(s) = \text{sign}\left(\sum_{k=1}^{N} W_{ik}^1 s_k\right)\right\} = \Pr\left\{\phi_i(s) = \text{sign}\left(\sum_{k=1}^{N} W_{ik}^2 s_k\right)\right\} \tag{24}$$

for Gaussian matrices $W^1$ and $W^2$ differing by inversion of a single element.

Unfortunately, the expectation, without any higher moments known, does not provide information about a typical case. Depending on the topology $T$, a typical realization of $W$ might have $M = O(1)$. Alternatively, there could be a few exceptional realizations with a huge number of fixed points while typically $M = 0$.

For example, for $T = I$ there are $M = 0$ fixed points with probability $1 - 2^{-N}$ and $M = 2^N$ with probability $2^{-N}$. Conversely if $T$ is a cyclic graph (e.g. cyclic permutation) then with probability of *half* the network has two fixed points, and it has *zero* fixed points in the complementary case. Although it is not immediately clear which scenario is relevant for scale free networks, for lumped hub networks some insights are available.

For a lumped hub network with a strong hub, such that $\nabla d < 1$ in the open loop setting, global convergence to a stable fixed point $s' = s^*$ is expected. If the consistency condition associated with relaxation of clamping $h(t)$ in Eq (5) is met, then the closed loop system inherits the fixed point at $s = s^*$. Moreover, another fixed point emerges at $s = -s^*$ corresponding to a drive of $-1$ in the open loop system. Since the aforementioned condition is met with probability *half* we have an expected number of $2 \times \frac{1}{2} = 1$ fixed points in agreement with [38].

In the opposite extreme case of *zero* drive from the hub, it follows from mean field theory that the network exhibits chaotic dynamics with any pair of state-space trajectories becoming asymptotically orthogonal. In particular, to comply with MFT, distinct fixed points should be either orthogonal to- or inverse of one another. We assume, without a rigorous proof, that orthogonality approximately implies independence. Namely, events of having a fixed point at $s = s^1$ and at $s = s^2$ are statistically independent for $s^1 \perp s^2$. Now, we recall that fixed points appear in pairs. Hence the number of *pairs* of fixed points is distributed binomially with parameter $n = \frac{2^N}{2} = 2^{N-1}$ of possible pairs $\pm s^*$ of a state and its inverse, and a probability $p = 2^{-N}$ that a particular pair of points is fixed. In the limit of $N \to \infty$ this distribution converges to Poisson with parameter $\lambda = pn = \frac{1}{2}$:

$$\frac{M}{2} \sim Poisson\left(\frac{1}{2}\right). \tag{25}$$

To validate this theory numerically, we performed an exhaustive search for fixed points in small, fully connected networks with $16 \le N \le 22$ nodes. For $N > 22$ an exhaustive search was not feasible due to computational constraints. The results, shown in Fig 8, depict an approximate match to the theory which is not perfect, with inaccuracies exceeding the standard error. We attribute these inaccuracies to the finite size of the networks, emphasizing that orthogonality of chaotic trajectories upon which our theory is built, is not achieved, even approximately, for values of $N$ small enough for exhaustive search. Future work may include numerical experiments with larger and sparser networks using advanced search algorithms e.g. the method of [46].

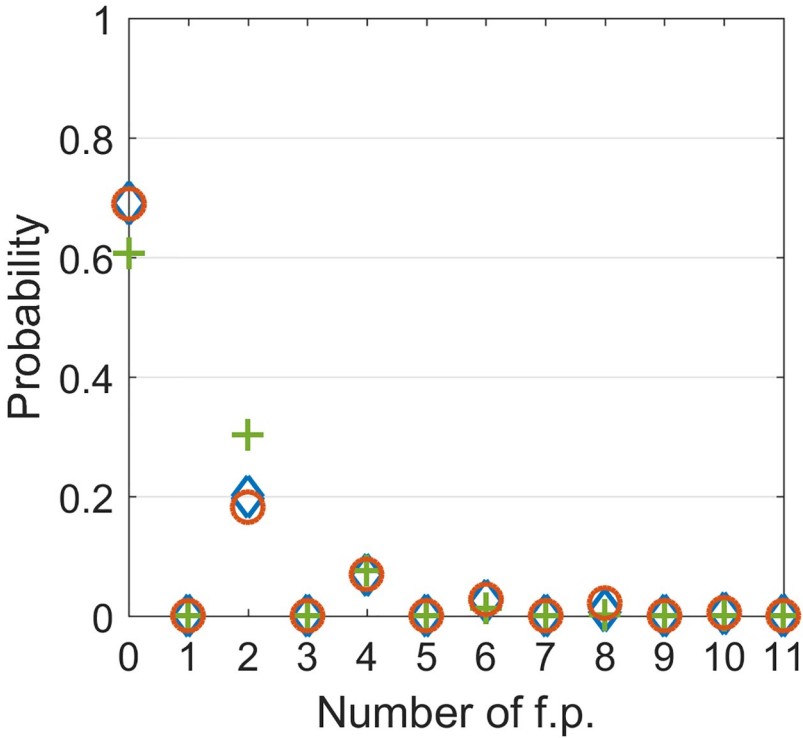

**Fig 8. Probability of number of fixed points in Boolean networks.** '+'—theoretical prediction. ∘ and ◇—empirical estimates for networks of size $N = 16$ and 19 based on $10^4$ and $10^3$ random network realizations respectively.

For intermediate values of $\sigma_h$ the picture is more complicated. Chaos is not suppressed but the symmetry between $\pm s$ is broken by the drive (in the open loop setting only, since once the loop is closed symmetry is restored). For $\sigma_h \ll 1$ one might repeat our reasoning for $\sigma_h = 0$, this time with single fixed points rather than pairs, and therefore with Eq (25) being replaced by:

$$M \sim Poisson(1). \tag{26}$$

Numerical observations on small networks reported in Table 1 tell us that while a sharp transition in the distribution of fixed points is observed once a drive is introduced, the distribution of $M$ does not immediately match Eq (26). Here again we tend to attribute such a discrepancy with theory to finite size effects compromising the fixed point orthogonality.

**Table 1. Numerical estimates for the distribution of $M$ are shown as a function of external drive strength for an ensemble of dense random networks of size $N = 16$ with Gaussian i.i.d elements.** Random Gaussian drive is scaled by $\sigma_h$. Poisson distribution with parameter $\lambda = 1, \frac{1}{2}$ are provided for reference.

| $\sigma_h$ | M: | 0 | 1 | 2 | 3 | 4 | 5 |
|---|---|---|---|---|---|---|---|
| 0 | | 0.679 | 0 | 0.206 | 0 | 0.069 | 0 |
| 0.1 | | 0.563 | 0.205 | 0.097 | 0.048 | 0.028 | 0.020 |
| 0.5 | | 0.505 | 0.248 | 0.117 | 0.062 | 0.027 | 0.025 |
| 1.0 | | 0.435 | 0.305 | 0.149 | 0.065 | 0.024 | 0.008 |
| 3.0 | | 0.279 | 0.481 | 0.185 | 0.043 | 0.009 | 0 |
| 10.0 | | 0.112 | 0.780 | 0.095 | 0.012 | 0.001 | 0 |
| $\lambda$ | x: | 0 | 1 | 2 | 3 | 4 | 5 |
| $\frac{1}{2}$ | | 0.607 | 0 | 0.303 | 0 | 0.076 | 0 |
| 1 | | 0.607 | 0.304 | 0.076 | 0.013 | 0.002 | $2 \times 10^{-4}$ |

To conclude, we present strong evidence, that the event of having a fixed point in the dynamical system (1) has probability of order *one* and it is the fixed point stabilities rather than their existence that determines the convergence probabilities of dynamical numeric simulations like in Fig 1.

Remarkably, this result is opposed to a case of continuous activation function (e.g. tanh) where an exponentially large amount of fixed points appear in the chaotic regime [47]. An in-depth analysis of these differences is beyond the scope of the current work.

## Estimation of hub flipping time in perturbed closed loop network

To obtain a, fairly coarse, lower bound on $T_{flip}$ one may argue as follows: the hub has approximately $k_h = m\langle k \rangle = O(1)$ incoming connections whose weighted sum determines its sign. Let $H_{in}$ denote the set of these nodes. Given a random perturbation of magnitude $\delta \ll 1$ to the fixed point $s^*$ the probability for a single node in $H_{in}$ to be flipped is

$$p_1 = \text{Pr} \ \{\text{single flip in } H_{in}\} = \delta \frac{k_h}{N} \ll 1 \tag{27}$$

and the probability of $q$ flips is $p_q \approx p_1^q = O(\delta^q N^{-q})$ which can be neglected for small $\delta$. Finally, by the arguments developed above, the probability that the hub will flip due to a single node flip is $\frac{2}{\pi} \arctan \frac{1}{\sqrt{k_h}}$ and consequently, the flip rate is given by:

$$\text{Pr} \ \{\text{hub flips}\} = \delta \frac{k_h}{N} \frac{2}{\pi} \ \arctan \ \frac{1}{\sqrt{k_h}}. \tag{28}$$

This corresponds to a typical flip time of:

$$T_{flip} \approx \frac{N}{\sqrt{k_h}\delta} \tag{29}$$

Hence, the stability condition Eq (10) clearly holds for $-\log \nabla d$ of order *one*.

Even in the critical regime, where $\log \nabla \to 0$ is approached, condition Eq (10) translates into

$$1 - \nabla d > \frac{\sqrt{k_h}\delta}{N} \ \log \ \delta N \tag{30}$$

which for our typical setting of $N = 1500$, $< k_h > \approx 20$, and a reasonable small perturbed fraction of $\delta = 0.1$ of the nodes translates into a requirement of:

$$1 - \nabla d < 1.5 \times 10^{-3}. \tag{31}$$

This is very close to criticality compared to other effects that may dominate inaccuracy of mean field calculations even in *open* loop setting. (Compare to Fig 5 that depicts open loop and closed loop setting, and Fig 6 which includes plot of convergence probability vs. $\nabla d$).

## Beyond convergence: Dynamical properties of lumped hub approximation

Most of the analysis in the main text focused on the probability of convergence to quasi fixed points. Here we demonstrate that even when the networks do not converge, some of their dynamical properties are still captured by the lumped hub approximation. First, we computed the participation ratio as a measure of the dimensionality of dynamics in the non-converging case. The correspondence between SFO and lumped-hub approximation is seen in Fig 9, similar to the match of convergence fraction shown in Fig 4.

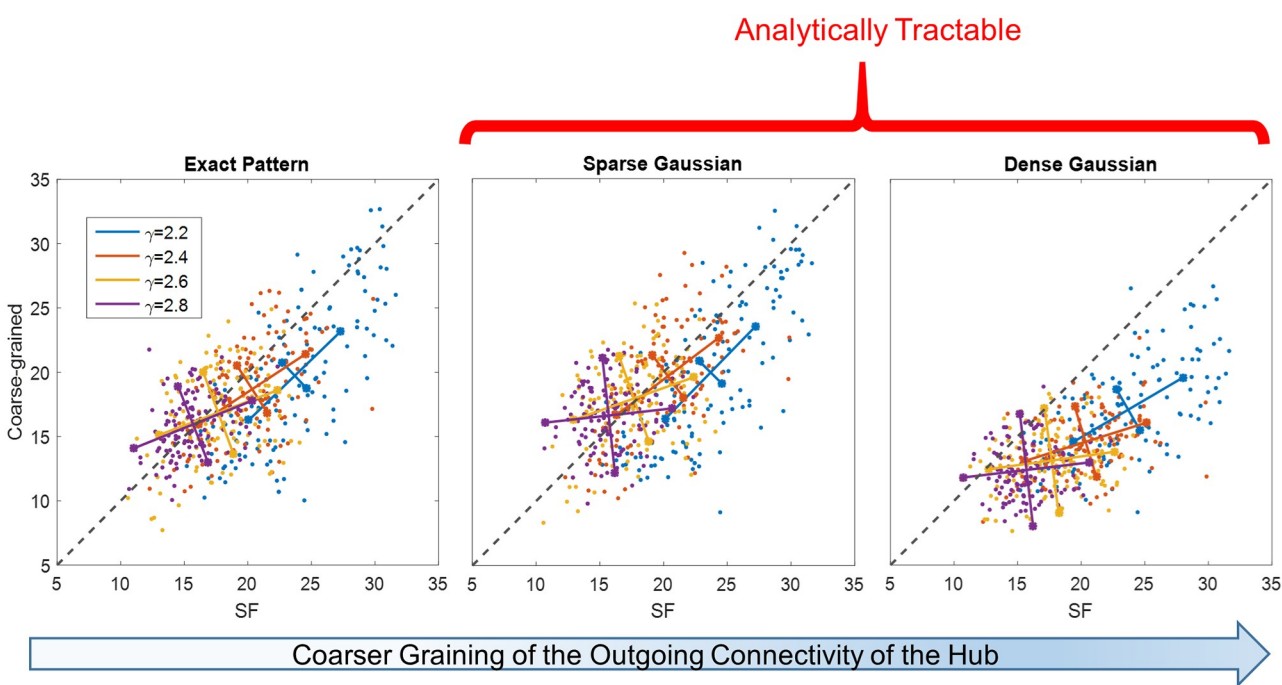

**Fig 9. Trajectory dimensionality captured by lumped hub approximation.** The participation ratio of trajectories, defined as $\frac{(trC)^2}{tr(C^2)}$ with $C$ the covariance matrix of the trajectories $s(t)$, is compared between scale free network and Lumped-hub approximations of increasingly coarser graining. Notations and methods are similar to Fig 4.

Second, we computed the autocorrelation functions of dynamic trajectories. Like other statistics, they greatly vary from one realization to the next in SFO ensembles. Nevertheless, average correlations functions—also, like other statistics—show a favorable comparison to the corresponding lumped-hub ensemble (Fig 10). This is quantified by the exponential fit to the autocorrelation function, after selecting only trajectories with low asymptotic autocorrelation.

## Constructing samples of heterogeneous network ensembles

To construct network samples from an ensemble with a SFO dgree distribution, the adjacency matrix is constructed by first randomly sampling a sequence of $N$ degrees from a scale-free distribution, and assigning a degree, $k$, from that sequence randomly to each node in the network. For each such node a set of $k$ random outgoing connection is chosen. This procedure results in a scale-free outgoing distribution and Binomial incoming distribution as there are no constraint on the incoming distribution and the choice of incoming degrees is purely random. SFI ensembles where constructed by transposing SFO networks created as described above.

The scale-free sequences are obtained by a discretization to the nearest integer of the continuous Pareto distribution $P(k) = \frac{(\gamma - 1)k_{min}^{(\gamma-1)}}{k^\gamma}$. This is implemented by the continuous MATLAB Generalized Pareto random number generators with Generalized Pareto parameters $k = 1/(\gamma - 1)$, $\sigma = k_{min}/(\gamma - 1)$ and $\theta = k_{min}$.

## Sign function

Networks with sparse connectivity often have entire rows that are zero; namely, nodes are created that receive zero input. With the definition Eq (2), such nodes retain their value of zero

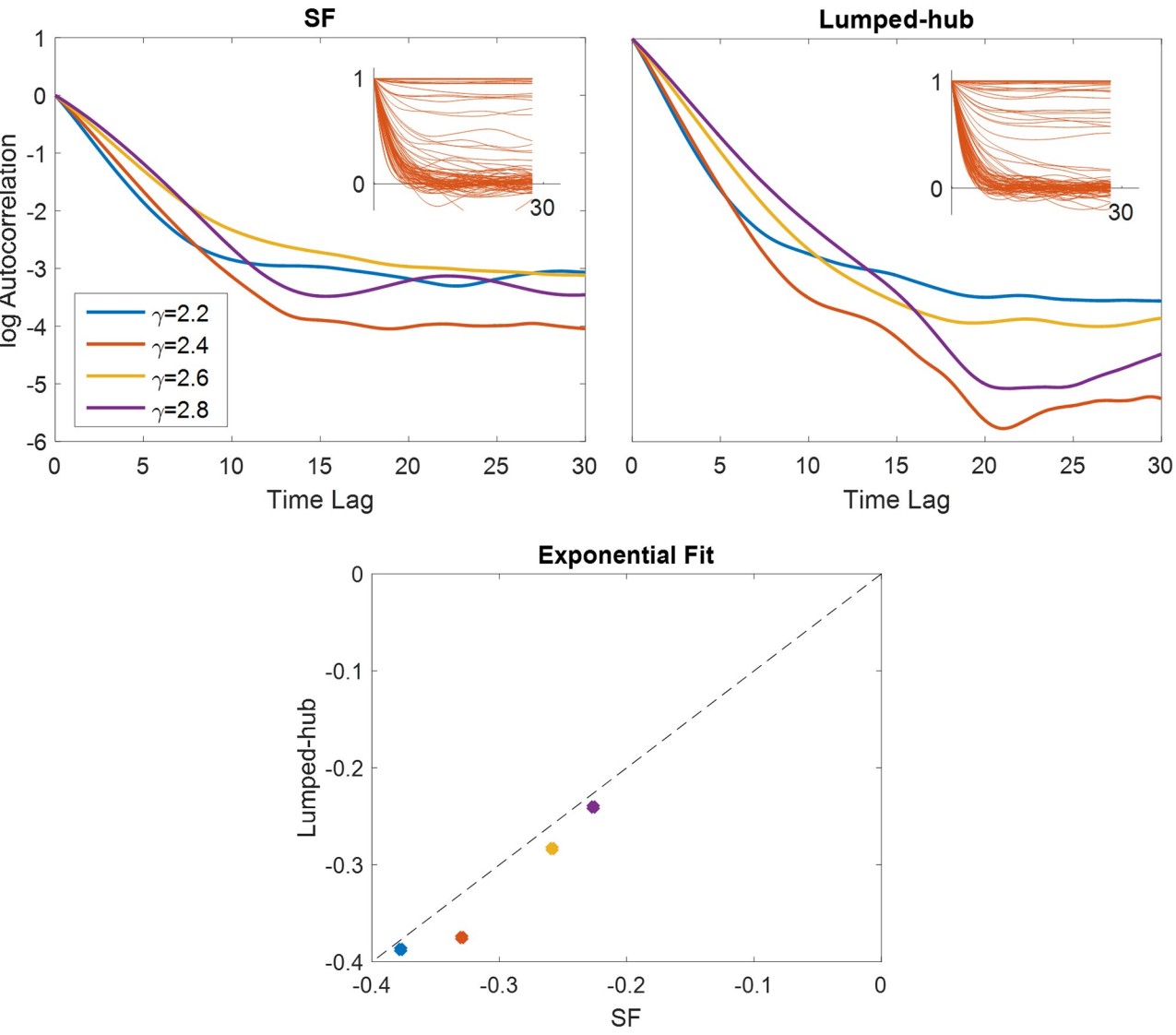

**Fig 10. Autocorrelation of SFO ensembles vs. their lumped hub approximation. Top**: Log of mean autocorrelation is shown for scale free (left) and corresponding lumped-hub (right) networks. The mean is obtained by averaging only over trajectories with asymptotic autocorrelation below 0.3. Insets show 100 individual trajectories obtained for $\gamma = 2.4$. **Bottom**: Exponential fits for the autocorrelation functions in the range $0 \leq t \leq 5$ ($\gamma$ color coded accordingly to the top panel.

and therefore effectively do not participate in the dynamics. The effective network thus has slightly modified size and statistical parameters. We have verified by simulation that all our results are weakly affected by this modification. In particular, the changes induced in $\nabla d$ are smaller than the finite-size effects.

Other definitions of the sign function, for example

$$\text{sign}(x) = \begin{cases} 1, & \text{if } x >= 0 \\ -1 & \text{if } x < 0, \end{cases} \tag{32}$$

do not effectively prune these source nodes. Instead, these source nodes become effectively equivalent to an external drive and thus increase the stability of the dynamics.

### Probability of convergence to fixed points and quasi fixed points (QFPs)

The networks' frozen cores were determined by running the dynamics for an initial convergence period of 4000 time steps, with 10% of networks nodes being updated at each step, and then measuring the fraction of nodes which were frozen within an additional time interval of 1000 steps (altogether 5000 time steps). Some simulations were run for double the time steps (a of total 10000 time steps, 2000 of which used for determining frozen core). No detectable difference was observed with such longer times. QFPs were defined as states with a frozen core larger than 90%, while fixed points are networks with a frozen core of 100%. The probability of convergence to a fixed point or QFP was calculated by simulating the dynamics of a an ensemble of 500 networks, unless stated otherwise, and measuring the fraction of networks in the ensemble which reached the relevant frozen core criterion.

## Author Contributions

**Conceptualization:** Alexander Rivkind, Hallel Schreier, Naama Brenner, Omri Barak.

**Formal analysis:** Alexander Rivkind, Hallel Schreier.

**Funding acquisition:** Naama Brenner, Omri Barak.

**Investigation:** Alexander Rivkind, Hallel Schreier.

**Methodology:** Alexander Rivkind, Hallel Schreier.

**Supervision:** Naama Brenner, Omri Barak.

**Writing – original draft:** Alexander Rivkind, Hallel Schreier, Naama Brenner, Omri Barak.

**Writing – review & editing:** Alexander Rivkind, Hallel Schreier, Naama Brenner, Omri Barak.

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
