## [Decision Letter · Decision Letter 0]

21 Nov 2019

Dear Dr Brenner,

Thank you very much for submitting your manuscript 'Scale free topology as an effective feedback system' for review by PLOS Computational Biology. Your manuscript has been fully evaluated by the PLOS Computational Biology editorial team and in this case also by independent peer reviewers. The reviewers appreciated the attention to an important problem, but raised some substantial concerns about the manuscript as it currently stands. While your manuscript cannot be accepted in its present form, we are willing to consider a revised version in which the issues raised by the reviewers have been adequately addressed. We cannot, of course, promise publication at that time.

Sincerely,

Alexandre V. Morozov, Ph.D.

Associate Editor

PLOS Computational Biology

Mark Alber

Deputy Editor

PLOS Computational Biology

[LINK]

Reviewer's Responses to Questions

**Comments to the Authors:**

Reviewer #1: Rivkind et al. submitted an interesting numerical study of the scale-free networks of interacting binary units (Boolean). The authors first show the convergence probability (over finite realizations) to a fixed point is much lower for Scale-Free-Out (SFO) compared with Scale-Free-In (SFI) degree networks. The second result of their paper is the illustration of "lumped hub approximation" for the SFO network. Remarkably, lumped hub approximation captures the distribution of frozen states and convergence to probability to a fixed point. After that, the authors analyze a Mean-Field Theory (MFT) for the lumped hub approximation to give more insight for the effect of hubs on the recurrent dynamics. Here, the authors study two closed and open-loop interactions between the hub and bulk part of the approximate network. Furthermore, they investigate the applicability of the MFT for the open/closed loop approximation.

I believe this is undoubtedly nice work, and it is relevant for the various subfields of computational biology and theoretical neuroscience. Therefore, I support its publication in PLoS CB.

However, in my opinion, the authors have to address three major points in the current manuscript to secure the desired impact:

First, it is unclear to me whether SFO networks converge to point-wise deterministic dynamics. The variance of the column sum in these networks is diverging, and as a result, it breaks the conditions for the existence of a deterministic MFT, as described in Farkhooi and Stannat (PRL, 2017). This has to be discussed in the revision and the notion of a fixed point in this context has to be appropriately defined.

Second, the comparison between the SFO and lumped hub approximation can be improved. I'd suggest two different comparisons in temporal statistics of the networks, and time-averaged properties should be given. This comparison can be used to motivate further analysis in the paper.

Third, the open/closed-loop analysis using MFT can be improved. It is not so clear for me (in the way that it is written in the main text) if suppression of chaos and the phase transition depends on the approximation itself. I wonder if authors could provide auto-correlations of the system before phase transition for both the approximation and full SFO networks.

Reviewer #2: The authors investigate how the topology determines the controllability of a dynamic system, and investigate specifically the impact of the high-degree nodes in a scale-free topology. Scale-free topologies are characterized by many nodes with only one or a few connections, some nodes with intermediate number of connections, and a few nodes with very many connections. They bundle the highly connected nodes into a single one, and replace the remaining part of the scale-free network by a random network. This simplification facilitates analytical mean-field treatment. They then show under which conditions the network converges – or does not converge to a fixed point.

I am divided with respect to the impact and generality of the results. On the one hand, it is a very elegant and creative framework, and the results are convincing. On the other hand, the approximations for the mean-field are rough, and the dynamics on the network is very simple, including the fixed-point as an absorbing state. Thus, I am not yet convinced that the framework is of general biological relevance.

Specific comments:

Major:

It remains a question whether the “lumped-hub” approximation is a good one. Taking e.g. figure 4A, there is some correlation, but I guess almost any coarse-graining of the topology would result in some correlation here. Can you argue that your choice of coarse-graining is a good one in general?

Specifically, why lumping together four, not three or five of the most connected nodes? Why randomizing the other topology, not just keeping it? How much does it impact the results?

Minor:

Abstract: “Based on the observation that in finite networks a small number of hubs have a

disproportionate effect on the entire system, we construct an approximation by lumping

these nodes into a single effective hub, which acts as a feedback loop with the rest of the

nodes.“

Only fairly late it becomes clear, why you combine the set of ‘most important nodes’ to one node, instead of averaging exactly in the opposite manner, namely over all the non-important hubs:

It seems to be much more intuitive, to reduce e.g. all the nodes with degree 1, which is the most frequent degree in a scale-free network, and then potentially also remove the other low-degree nodes, until a treatable topology is found.

It would be good to make the aim more clear at an earlier point.

The terms “source” and “sink” might help the description for the outgoing/ incoming hubs.

Please discuss whether scale-free distributions are really a reasonable description for neural topology: Is it really the case that the most frequent projection pattern of a neuron is to connect to just one other neuron? (p(k=1)>=p(k=N) for all N. – I doubt so.

Equation 1: I’d suggest to write out the sign function.

Figure 4: I’d suggest to use CDFs instead, and plot both PDFs in a single panel.

Fig. 5: “Mean field theory predicts a phase transition at \\sigma_crit”.

Could you elaborate about the position of the \\sigma_crit? I would have expected it at about \\sigma_h=1.2 for the red, and close to zero for the blue condition, i.e. at the point where QFPs start to emerge?

**Have all data underlying the figures and results presented in the manuscript been provided?**

Reviewer #1: Yes

Reviewer #2: No:

PLOS authors have the option to publish the peer review history of their article (what does this mean?). If published, this will include your full peer review and any attached files.

Reviewer #1: No

Reviewer #2: No

---

## [Decision Letter · Decision Letter 1]

26 Mar 2020

Dear Professor Brenner,

We are pleased to inform you that your manuscript 'Scale free topology as an effective feedback system' has been provisionally accepted for publication in PLOS Computational Biology.

Best regards,

Alexandre V. Morozov, Ph.D.

Associate Editor

PLOS Computational Biology

Mark Alber

Deputy Editor

PLOS Computational Biology

Reviewer's Responses to Questions

**Comments to the Authors:**

Reviewer #1: At first, I apologize for my delay in responding to the revision of the submitted manuscript. I was rather ill in the last three weeks.

I read the revision and the authors response completely and now I am confident that their work is interesting and very relevant for the readership of PLoS CB. In their version, they provide convening additional results on the temporal dynamics of the network activities. Also, they revised the text to be more accessible to readers.

Therefore, I recommend the revision for publication in PLOS CB.

**Have all data underlying the figures and results presented in the manuscript been provided?**

Reviewer #1: Yes

PLOS authors have the option to publish the peer review history of their article (what does this mean?). If published, this will include your full peer review and any attached files.

Reviewer #1: No

---

## [Editor Report · Acceptance letter]

4 May 2020

PCOMPBIOL-D-19-01483R1 

Scale free topology as an effective feedback system

Dear Dr Brenner,

I am pleased to inform you that your manuscript has been formally accepted for publication in PLOS Computational Biology. Your manuscript is now with our production department and you will be notified of the publication date in due course.

With kind regards,

Bailey Hanna
